# Advances in Omic Studies Drive Discoveries in the Biology of Anisakid Nematodes

**DOI:** 10.3390/genes11070801

**Published:** 2020-07-15

**Authors:** Stefano D’Amelio, Fabrizio Lombardo, Antonella Pizzarelli, Ilaria Bellini, Serena Cavallero

**Affiliations:** Department of Public Health and Infectious Diseases, Sapienza University of Rome, 00185 Rome, Italy; stefano.damelio@uniroma1.it (S.D.); fabrizio.lombardo@uniroma1.it (F.L.); antonella.pizzarelli@uniroma1.it (A.P.); ilaria.bellini@uniroma1.it (I.B.)

**Keywords:** anisakids, genomic, transcriptomic, proteomic

## Abstract

Advancements in technologies employed in high-throughput next-generation sequencing (NGS) methods are supporting the spread of studies that, combined with advances in computational biology and bioinformatics, have greatly accelerated discoveries within basic and biomedical research for many parasitic diseases. Here, we review the most updated “omic” studies performed on anisakid nematodes, a family of marine parasites that are causative agents of the fish-borne zoonosis known as anisakiasis or anisakidosis. Few deposited data on *Anisakis* genomes are so far available, and this still hinders the deep and highly accurate characterization of biological aspects of interest, even as several transcriptomic and proteomic studies are becoming available. These have been aimed at discovering and characterizing molecules specific to peculiar developmental parasitic stages or tissues, as well as transcripts with pathogenic potential as toxins and allergens, with a broad relevance for a better understanding of host–pathogen relationships and for the development of reliable diagnostic tools.

## 1. Introduction

Anisakids (of the superfamily Ascaridoidea) are cosmopolitan parasitic nematodes that depend on aquatic hosts to successfully complete their life-cycle. Definitive and intermediate hosts are marine mammals or fish-eating birds and crustaceans, respectively, while fish and cephalopods can act as paratenic hosts [1]. The adults live in the gastrointestinal portion of the final hosts, which release the eggs into the sea through feces. After molts, embryonated eggs with third-stage larvae (L3) are ingested by intermediate hosts, which, in turn, are ingested by fish or cephalopods; they bore through their digestive tract wall and pass into the visceral body cavity, where the host-induced encapsulation take place; then, infected paratenic hosts are in turn ingested by the final host [2]. 

Anisakids are responsible for a relatively poorly known food-borne zoonosis named anisakiasis or anisakidosis that results from the accidental ingestion of marine products harboring infective L3 in the edible parts. The pathology shows mild-to-severe gastrointestinal and/or allergic symptoms and clinical signs, such as rhinitis, urticaria, and anaphylactic shock [3], and there is increasing evidence that suggests the occurrence of tumors in the same localization of larvae [4,5,6]. For these reasons, the need to fill the gap in knowledge about the pathogenetic mechanisms of such parasitic disease appears clear.

Incidence is strictly related to the tradition food habit of eating raw fish. Indeed, Japan alone accounts for around 90% of the total reported cases, with the remaining occurring in other countries such as Korea, China, Peru, Netherlands, Germany, France, Spain, Croatia, and Italy [7]. This food-borne zoonosis is now considered to be emerging in Europe [8], recently ranked as fifth in the European risk ranking and as second of 24 food-borne parasitosis with the highest “increasing illness potential” [9]. 

The taxonomy and phylogeny of anisakid nematodes have been widely studied in the past few decades (see Box 1 and Figure 1), with the adult and larval stages being studied and thus allowing for the description of a clear picture of the wide diversity of existing sibling and morphospecies [10,11]. The evolutionary relationships display a complex scenario, where marine Raphidascarididae appear more closely related to terrestrial Ascarididae than to Anisakidae, the latter being paraphyletic [12,13]. The high genetic variation that has been observed could be responsible for the parasites’ remarkable ability to adapt to different climatic regions and to several host species [14].

Given the presence of anisakids in all regions and in numerous animal hosts, their occurrence in the marine environment and fish products should be considered as a natural event and not as a contamination. Prevention and control strategies should therefore be based on the continuous monitoring of food-hazard and on social education [8,15]. 

In addition to the utility of anisakids as biological indicators for monitoring marine biodiversity and related trophic webs, they may also represent a good model to study mechanisms of pathogenesis and host-manipulation strategies in parasitic diseases. In fact, the collection of alive infective L3 is relatively easy, given their large occurrence in fish that are available in market places. Moreover, their simple maintenance in basic culture media allows for the testing of several features of interest for basic research, as well as for applications in diagnosis and therapeutics. Such aspects are nowadays sustained by advancements in technologies aimed at describing the molecular profiles of the studied organisms and at discovering their widest levels of diversity. In this context, the employment of high-throughput next-generation sequencing (NGS) methods is greatly increasing. In the last decade, such approaches, combined with advances in computational biology and bioinformatics, have accelerated discoveries within basic and biomedical research for many parasitic diseases [16,17]. 

On the other hand, the genomic and transcriptomic profiling of anisakids by the use of advanced molecular approaches are still scarce. Moreover, the first studies encountered unexpected bioinformatic complexities. Indeed, there have been examples wherein the genomes of parasitic helminths are more complex than that of their closely related models or free-living organisms [18,19,20]. Several efforts have been made to improve the genomic studies of parasitic nematodes, starting from the first draft genome of *Brugia malayi* [21] published in 2007. To date, 134 genome datasets of parasitic nematodes (completed, drafted, or partial) are available in WormBase Parasite [22], revealing the massive impact of modern technologies in this field in the last decade. Similarly, the implementation of RNA-seq technology has rapidly improved the expression databases based on ESTs (Expressed Sequence Tag) libraries and microarray analyses; see, as an example, [23,24,25]).

The current high-throughput technologies and the associated bioinformatics therefore represent new powerful tools to investigate genes and gene products in anisakids that are involved in fundamental biological mechanisms such as host–parasite interactions and pathogenesis, evolution and speciation, and the sensitization and characterization of immunoreactive proteins. Improving molecular knowledge on this family of parasites may be helpful in several ways, spanning from basic research to clinical and public health applications. These are mostly related to: (i) food-safety, given the presence of infective larvae in food products [26,27]; (ii) allergy-mediated diseases, even in relation to work exposure or to potential therapeutic use of parasitic derived products [28]; and (iii) the understanding of carcinogenic processes in relation to nematode infection [29].

Here, we review the most recent studies carried out using modern approaches aimed to investigate genomes, transcriptomes, and proteomes of anisakid nematodes, with particular attention to anisakidosis, from allergens to pathogenic parasitic behavior. These studies have provided novel and fundamental resources for biological and biotechnological research to understand the biology of these species and related nematodes. 

Box 1Pillars from the past.By the use of molecular markers (nuclear and mitochondrial regions), nine species belonging to three evolutionary phylogenetic lineages and one hybrid form have been described in the *Anisakis* genus (*Anisakis simplex* sensu *Anisakis*
*ziphidarum*, *Anisakis*
*nascettii*, *Anisakis*
*paggiae*, *Anisakis*
*brevispiculata*, *Anisakis*
*physeteris*, and *Anisakis*
*typica*). Among these taxa, only two have been commonly associated to anisakiasis so far: *A. simplex* s.s. and *A. pegreffii* (see also Figure 1). In vivo and in vitro studies on these zoonotic species have suggested a differential pathogenic potential, revealing a different ability to penetrate synthetic or
animal tissues or the propensity to trigger allergic reactions.

## 2. Advances in the Genomics of Anisakid Species

Parasitic nematodes are masters of host manipulation, and infections are rarely lethal but typically chronic, leading to pain, malnutrition, and disabilities. Despite their global distribution, their occurrence is mostly tropical, and little economic efforts have been made in the past to study treatments and control strategies. However, the genomic resources required for research into parasitic nematodes are still scarce. In the last two decades, the World Health Organization (WHO) has focused on global strategies to control morbidity due to infections with soil-transmitted helminths and listing them among the neglected tropical diseases (NTDs [30]). In a complementary vision, the International Helminth Genomes Consortium (2019) recently performed the largest genome comparison of parasitic and non-parasitic worms, combining 36 published genomes. The study had the ambition to identify gene families and processes associated with the major parasitic groups, accelerating the search for new interventions through mining the data set of more than 1.4 million genes. Such data may have numerous applications in functional genomics, e.g., designing probes and identifying putative protein sequences in proteomics, helping to predict new drug targets and drugs. Among anisakids, the draft genome of the species *Anisakis simplex* was the first available in the website Wormbase ParaSite [22], elaborated by the Parasite Genomic group at the Wellcome Trust Sanger Institute, in the framework of the 50 Helminth Genomes project (PRJEB496 project). It counts 126,869,778 bp and a 20,971 predicted genes. In the same year, another laboratory sequenced the *A. simplex sensu lato* (*s.l.*) genome from herrings of the Baltic Sea region, obtaining a total of 26,479,344 reads and an average of 65,000 contigs (while 49,180 are reported in the PRJEB496 project). The aim of this study was to analyze mainly the carbohydrate metabolism in L3 and L4 *A. simplex s.l.* developmental stages [31], with a particular emphasis on glycogen and trehalose, the most important saccharides in the life cycle of parasitic nematodes because they are involved in molting processes and stress response. 

Results suggested shared features with other parasitic nematodes, such as the mean coding sequence length (*A. simplex s.l*.: 1234 bp, compared to *Toxocara canis*: 1156 bp), the average gene length shared by members of the parasitic clade III (*A. simplex*: 429 bp; *Brugia malayi*: 434 bp), and the high value of the mean intron length (*A. simplex* s.l.: 286.7 bp). The study confirmed previous evidence obtained on *A. simplex* sensu stricto (*s.s.*) and *Anisakis pegreffii*, as well as in other parasitic nematodes (namely *Haemonchus contortus*), on amino acid and carbohydrate metabolism physiology and transport [32,33]. A possible relationship of sugar metabolism with pathogenicity is also suggested, as it is the main pathway in nematode physiology [34].

Despite the fact that genomic studies are increasing, the only available source of information about the *A. simplex s.l.* genome are scaffolds and raw data in GenBank and Wormbase ParaSite. The *A. simplex* complex includes three species (namely *A. simplex sensu stricto*, *A. pegreffii,* and *Anisakis berlandi*) and one hybrid form, which may complicate the precise definition of genomes, transcripts, and proteins, as well as the precise assignment of genes or gene products to a given species. Dedicated and detailed assays for species of interest should also be done to overcome issues associated with previous genomic studies based mostly on homology to *Caenorhabditis elegans*, a phylogenetically distant organism, in the absence of more suitable reference genomes, represented by the terrestrial related species *Ascaris suum* [35]. Moreover, nine species are described to date in the *Anisakis* genus, eight are described to date in the *Pseudoterranova* genus, and many others are described to date in the *Contracaecum* genus. Among all, only few species have been reported in human infections. This has been often related to peculiar abilities that lead to pathogenicity, but other factors should be considered, such as their occurrence in fishes that are commonly eaten raw, or distribution in countries where the molecular diagnosis of human cases is well established in research and diagnostic laboratories. Only two studies investigated the pathogenic potential of species other than *A. simplex* and *A. pegreffii*, revealing that *Anisakis physeteris* and *Anisakis paggiae* could be capable, to a lesser extent, of attaching to and penetrating the gastrointestinal wall of model animals [36] and the ability of *Contracaecum osculatum* to elicit a granulomatous reaction at the penetration site in pig gastrointestinal mucosa [37]. 

Studies based on genome-wide approaches have provided clues to understand additional biological features of anisakids, such as the amount of genetic variation, the adaptive potential, hybridization, and introgression events. Thus, DNA microsatellites have been used as alternative nuclear markers for the species recognition and population genetics of *A. simplex s. s.*, *A. pegreffii*, and *A. berlandi* [38,39,40]. Similarly, the sequencing of the entire mtDNA may help in performing more accurate phylogenetic inferences to depict evolutionary relationships of species and testing hypotheses. Entire mt circular genomes are only available for representative species of the three genera with pathogenic potential, namely of *Anisakis simplex s.s.*, *C. osculatum sensu stricto*, and *Pseudoterranova azarasi* [41,42], counting for 13,926, 13,823, and 13,954 bp in size, respectively. Phylogenetic analyses based on concatenated amino acid sequences showed that *Pseudoterranova* were more closely related to *Anisakis* than to *Contracaecum* [42]. In the future, phylogenomics will provide a more reliable phylogeny of anisakids.

## 3. Transcriptomic Analyses: Studies on Specific Developmental Stages and Tissues

Investigations of genes and gene products have been putatively involved in fundamental parasitic biological functions for nematodes, such as different molting stages, specific metabolic routes (e.g., carbohydrate metabolism), responses to stress (e.g., temperature), or interactions with the hosts, may assist the discovery of fundamental aspects related to physiological pathways that are potentially targeted by pharmacological therapies or the discovery of new molecules useful for diagnostic improvements. Studies aimed at characterizing the expression profiles of genes with known antigenic properties in different conditions or in different parasitic tissues are becoming available also for anisakids.

Comparative RNA-seq analyses on different developmental stages can help in understanding adaptive processes, such as molecular pathways linked to parasitic survival and discovery of stage-enriched gene expression. Kim et al. [43] performed such analyses on *A. simplex* L3 and L4. Several protease-related and protein biosynthesis-related genes were highly expressed in the infective L3, all of which are crucial for invading host tissues. Collagen synthesis-related genes were highly expressed in L4, suggesting the active biosynthesis of collagen during molting process. The same authors have recently published related studies based on the RNA-seq and de novo assembly of *A. pegreffii* L3 and L4 transcriptome profiles [44].

Temperature is a recognized factor that is implicated in adaptive processes in aquatic parasites, which perform larval molting or hatching in a temperature-dependent way and are induced to hypometabolism by cold temperatures [45]. Three proteins with antigenic roles among excretory secretory products, in particular the antigens A.peg-1 (Kunitz serine protease inhibitor), A.peg-7 (armadillo-like helical), and A.peg-13 (globin), have been studied in *A. pegreffii* via the evaluation of expression profiles under controlled temperatures simulating homeothermic and ectothermic hosts conditions [46]. While the A.peg-7 expression profiles did not significantly change among the different temperature conditions, A.peg-1 and A.peg-13 expression profiles were significantly higher at 20 °C and 37 °C, respectively. 

Carbohydrate components have been also studied in relation to different temperatures as triggers for stress, and *A. simplex s.s.* is able to synthetize trehalose both at low (0 °C) and high temperatures (45 °C), thus suggesting a fundamental role in providing energy during the thermotolerance and starvation processes. Sugars such as trehalose and glycogen are instrumental for survival under environmental stress conditions, and several studies recently investigated such aspects in zoonotic anisakids, also in relation to different developmental stages [31,47]. 

Aspects related to temperature may explain also the reduced pathogenic potential of *Hysterothylacium* spp. (Raphidascarididae family), a genus of marine parasites infecting teleosts at their adult stage. Despite their wide distribution, they have been rarely associated with human infections, and their pathogenic potential is scarce or negligible [48]. A possible explanation may be the strong cellular response to stress (thermic and oxidative) that *Hysterothylacium* species may produce. Heat shock or other stress, such as the accumulation of denatured proteins or hormone-like proteins, may be implicated in this behavior: A study on the in vitro cultivation of *Hysterothylacium aduncum* from L3 to egg-laying adults revealed that the best survival rates and percentage of molting to L4 occurred at 13 °C, while nematodes survived only for few hours at 37 °C [49]. 

Previous studies have investigated the role of hydrolases in *H*. *aduncum* using classic approaches reporting no trypsin- and chymotrypsin-specific—nor alpha-galactosidase-specific, alpha-mannosidase-specific, and beta-glucuronidase-specific—activities, thus showing that the role of hydrolases depends more on the type of host’s habitat than on the parasite’s taxonomic affiliation [50]. 

Similarly, enzymatic activities were investigated in excretion–secretion (ES) products and extracts of another marine parasite belonging to the Anisakidae family, *Contracaecum rudolphii* [51]. A high activity was revealed for esterase, alpha-glucosidase, leucine arylamidase, and valine arylamidase. Altogether, such evidence suggested that the ES products of larval or adult developmental stages contain active hydrolases that may damage the host’s alimentary tract epithelium. Moreover, the activities of almost all glucosidases in parasite’s extracts suggested carbohydrate metabolism as the main energy source.

Besides temperature and enzymatic activity, parasitic transcripts potentially involved in pathogenic behavior are of great interest because they are still obscure. In fact, a recent transcriptomic analysis performed on *Anisakis* species of medical relevance using a comparative approach between the whole body of the L3 vs the isolated pharyngeal region suggested that several gene products potentially involved in pathogenesis are enriched in that tissue [33]. Transcripts encoding for proteolytic enzymes, molecules encoding anesthetics, inhibitors of primary hemostasis and virulence factors, anticoagulants, and immunomodulatory peptides were also found to be enriched in pharyngeal tissues. This study produced a bulk of data that represent a ready-to-use resource for future functional studies of biological pathways specifically involved in host–parasite interplay. Functional annotations of parasitic excretory cells may help to clarify the pathogenesis of anisakiasis, because ES products are crucial for parasite infectivity and host immunomodulation. Several aspects of pathogenic behavior are related to larvae’s ability to penetrate host tissue in the gastrointestinal region and even in other locations [52,53], realizing an erratic migration that may cause both an acute response with an associated allergic reaction [54], or a secondary reaction associated with granuloma. Among congeneric species, *A. simplex s.s.* and *A. pegreffii* are the two main etiological agents of anisakiasis [55,56]. In vivo and in vitro studies on their pathogenic behavior have recently pointed out differential abilities of penetrating the muscular tissue of fish and surviving in artificial gastric fluids for these species and their hybrid form [55,57,58,59]. Differences also emerged in the transcriptomic repertoire of *A. simplex s.s.* and *A. pegreffii*, which confirmed transcripts that encode metalloproteinases (e.g., aminopeptidases, astacins, and neprilysins) as particularly abundant in *A. simplex s.s.* Additionally, other transcripts—such as those encoding for factors containing Kunitz domains, with trypsin inhibitor activity, and aspartic proteases—were found to be exclusively upregulated in the *A. simplex s.s.* pharyngeal region [33]. Differences among sibling species were also investigated in hybrid forms, and the results revealed strong parent-of-origin effects in hybrid transcripts repertoire [60]. Such evidence is of great interest in the study of nematode speciation events, as well as for the characterization of allergens. Llorens et al. [60] provided a bioinformatic platform capable of integrating all the data generated in the study to support the management of information on the evolutionary history and biology of *Anisakis* and related nematodes. In the *Anisakis* database [61], info on transcriptomes, secretomes, and allergomes are available for the two sibling species of medical interest and their hybrid forms. In the allergome file, 936 putative allergens are listed, providing a fundamental resource for further experimental testing.

## 4. Anisakids Proteomic Profiling

Proteins are fundamental molecules involved in several aspects of the parasite–host interface, and the importance of ES products for parasite survival in inhospitable environments is indisputable. Considering the natural role of these proteins, ES enzymatic molecules play multiple roles: They facilitate the hatching and molting of larvae, enable a parasite to migrate within tissues, inhibit blood coagulation, defend the parasite from host’s immunological response, and enhance feeding and nutrition. In fact, exploring the immunobiology of anisakid infections mainly relies on components of the ES parasitic products, and it represents an opportunity for the discovery and the development of new therapeutic approaches to deal with other gastrointestinal nematodes. Moreover, a detailed characterization of ES products may assist the development of specific and sensitive diagnostic tools, as well as the detection of parasitic signatures in food. In this scenario, the recent discovery of extracellular vesicles (EVs) secreted by helminths has revealed a new paradigm in the study of host–parasite relationships [62,63,64], adding complexity because helminth EVs have immunomodulatory effects and may contribute to pathogenesis, even promoting tumorigenesis, as has been demonstrated for parasitic flatworms [65]. 

The first proteomic study investigated aspects of medical concerns, comparing the protein profiles of the pathogenic *A. simplex* complex (*A. simplex s.s.*, *A. pegreffii*, and their hybrid) through 2D gel electrophoresis hybridized with pools of sera from *Anisakis* allergic patients and a parallel Western blot [66]. The Mediterranean species *A. pegreffii* is recognized as the main causative agent of anisakiasis in the European Mediterranean, such as Italy, Spain, and Croatia, and also in eastern countries such as Korea and Japan, where the presence of *A. simplex s.s*. has also been reported, though with a reduced involvement in human cases [56,67]. Thanks to MALDI-TOF/TOF analysis, differentially expressed proteins for *A. simplex s.s*., *A. pegreffii*, and their hybrid were described: Twenty-eight different allergenic proteins were classified, and most of them were described for the first time as potential new allergens in *Anisakis*. 

A similar approach was used by Fæste et al. [68] in a study based on gel banding patterns and IgE-immunostaining (IgE:immunoglobulin E) using sera from *A. simplex*-sensitized patients and proteome data obtained by mass spectrometry. Results showed that *A. simplex* proteins were homologous to allergens already characterized in other nematodes, insects, and shellfishes, which may be the source of possible cross-reactivity, as also confirmed by comparative genomic approaches [69]. This is a limiting factor for diagnostic specificity, as common phylogenetic routes or convergent evolution may lead to common protein solutions for similar pathways (e.g., the ability to molt shared by arthropods and nematodes, both belonging to the Ecdysozoa superphylum). 

Protein characterization may support methods for the fast monitoring of these parasites. Carrera et al. [70] used a three-step strategy based on the purification of anisakids’ thermostable proteins, their digestion with trypsin, and a final monitoring of several peptide markers by parallel reaction monitoring (PRM) mass spectrometry. Such a methodology may facilitate testing for regulatory and safety applications if integrated in defined control measures.

Recently, Stryiński and colleagues analyzed the global proteome of *A. simplex* L3 and L4 for the first time using tandem mass tag (TMT)-based quantitative proteomics [71]. The integrated strategy allowed them to detect sets of modulated proteins that provided a specific proteomic signature: L3 were characterized by pseudocoelomic globin, endochitinase 1, and paramyosin, and L4 showed neprilysin-2, glutamate dehydrogenase, and aminopeptidase N.

## 5. Characterization of Allergens

Omics-based techniques have been limitedly used in the framework of allergen characterization in anisakid nematodes despite significant advances in the pathobiology of allergic anisakiasis. Thus far, the exact number and nature of potential *Anisakis* spp. allergens remains unknown, and the characterization of such proteins is necessary for the development of robust and reliable diagnostic tools for allergic anisakiasis, which, in turn, underpins the implementation of effective therapeutic strategies.

Three kind of allergic anisakiasis are known: One is gastro-allergic anisakiasis, characterized by a combination of gastric symptoms and clinical signs together with allergic reactions; a second one is a classic allergic reaction, showing mild-to-severe symptoms and clinical outcomes (e.g., urticaria, rhino-conjunctivitis, asthma, dermatitis, and anaphylaxis). Allergic reactions are related not only to ingestion of infected food but also to inhalation or direct contact in the domestic or occupational environments where marine products are available.

A third form of allergic anisakiasis was suggested in relation to sensitized healthy individuals or asymptomatic patients, with a high anti-Anisakis IgE titer that potentially underwent a subclinical or undiagnosed gastric anisakiasis without allergic symptoms [72]. 

To date, around 14 allergens have been described (Table 1), and most of them have been detected in the parasitic somatic and ES products, with Ani s 1, Ani s 5, Ani s 7, and Ani s 9 being recognized by serum antibodies in the majority of individuals affected by allergic anisakiasis. Several allergens such as Ani s 1, Ani s 4, Ani s 5, Ani s 8, Ani s 9, Ani s 10, and Ani s 11 have shown heat stability [73,74,75], and this may cause a problem even when the infected fish is eaten cooked.

Ani s 1 is a major *Anisakis* allergen, and it belongs to the animal Kunitz serine protease inhibitor family (MEROPS group which includes peptidases and their inhibitors), showing two different isoforms: the most commonly recognized of 24 kDa and the second of 21 kDa [76]. This allergen was investigated by Morris and Sakanari [77], and it was then isolated and characterized in parasitic excretory glands by Moneo et al. [78], revealing that the serine protease inhibitor activity inhibited trypsin and elastase but not chymotrypsin. 

Ani s 2 or paramyosin is a 100 kDa highly conserved protein localized in invertebrate muscles. It shows a high frequency (88%) of IgE binding sites, and it has been suggested as strongly immunogenic in other helminthic parasitic infections caused by *Taenia, Schistostoma, Dirofilaria,* and *Onchocerca* [79]. The *Anisakis simplex* genome shows two genes coding for paramyosin, and the presence of a secreted form of Ani s 2 acting as a host immune response modulator as in *Taenia solium* is plausible [79]. 

Tropomyosin or Ani s 3 is also a pan-allergen, showing a significant homology with several known allergens including paramyosin [80], which is also shared by other organisms such as dust mites and crustaceans and which is potentially responsible for cross-reactivity. Ani s 3 is a 41 kDa protein that is able to mediate the interaction between the troponin complex in the striated muscular actin-linked calcium regulatory system [81]. 

Ani s 4 is a 9 kDa protein belonging to the cystatin family, as revealed by sequence analyses [82]. A recombinant form of 115 amino acids with a weight of 12,7 kDa was also observed. The first cystatin identified in nematodes that causes allergy in humans [83] is located in the excretory gland and underneath the cuticle. It can inhibit papain, a cysteine protease cleaving its substrate, and activate basophils from *A. simplex*- sensitized patients.

Ani s 5 is a 15 kDa protein of 152 amino acids identified in the crude extract of *A. simplex*, structurally highly homologous with nematode SXP/RAL-2 protein family. Identified for the first time with Ani s 6 by immune-screening [84], the latter is a serine protease inhibitor characterized by 84 residues that is able to inhibit alfa-chymotrypsin but not trypsin. Both allergens are secreted and may stimulate immune response in humans.

Ani s 7 is a major ES allergen of 139 kDa, as it is the only one identified in 100% of infected patients. Rodriguez et al. [85] characterized the structure of Ani s 7 for the first time, revealing a novel CX17–25CX9–22CX8CX6 tandem repeat peculiar motif. Ani s 7’s allergenic potential is related to the parasitic larval status: During the acute phase of infection, the immune system of rats reacts only if the larvae are alive [86].

Ani s 8 is a heat-stable protein of 15 kDa that was purified for the first time by Kobayashi et al. [87]. A member of the SXP/RAL-2 protein family, it shows cross reactivity with Ani s 5. These proteins are also involved in other parasite infections (e.g., in *A. suum* infections) that lead to IgE production. Similarly, Ani s 9 is a 15 kDa heat-stable protein and a member of the SXP/RA-L2 protein family, showing a 60% amino acid identity with the As-14 *A. suum* allergen. It is considered a minor allergen that shows 34% and 35% identity with Ani s 5 and Ani s 8, respectively, and analyses on human sera have reported cross-reactions between them [83,87].

Ani s 10 is a heat-stable protein of 21 kDa having no homology with any other described protein [88]. Ani s 11 (307 amino acids), the Ani s 11-like protein (160 residues), and Ani s 12 (295 residues) were characterized in 2011 [89]. With five or six types of short repetitive sequences comprising 6–15 amino acids, Ani s 11 and the Ani s 11-like protein share a 78% sequence identity. Ani s 12 has a longer tandem repeat structure (cysteine C and other residues X: CX13–25CX9CX7–8CX6, comprising 40–52 amino acid residues), and Ani s 11 and the Ani s 11-like protein are recognized by about half of *Anisakis* allergic patients.

The major allergen Ani s 13 is a hemoglobin protein with a theoretical molecular weight of 36.7 kDa. *Anisakis* hemoglobin was recognized by serum IgE in more than 50% of patients (64.3%) with cutaneous reactions after *Anisakis* parasitism [90]. The epitopes 2 and 5 seem to be relevant for IgE binding affinity and specificity. 

Ani s 14 is a 23.5 kDa protein characterized by two homologous regions with a common structure of CX8CX6CX22,26C and a 36% internal sequence identity [91]. This peculiarity was observed in other two *A. simplex* allergens (Ani s 7 and Ani s 12).

In the past, protein separation techniques were used to characterize some allergens, such as using HPLC to purify Ani s 1, Ani s 4, and Ani s 8 from the crude extract of the parasite [78,83,84]. The allergens identified thus far have been tested on sera from patients diagnosed with anisakiasis through specific immunoglobulin research. Immunoblotting is the technique commonly associated with more innovative screening techniques such as ELISA (enzyme-linked immunosorbent assay) and BAT (basophil activation test). Unfortunately, not all identified allergens have a high sensitivity: in fact, Ani s 14 and Ani s 12 showed a lower rate of reactivity to patients’ sera than Ani s 1, Ani s 2, Ani s 7, and Ani s 13, which are recognized by about 80% of patients or more [91]. Ani s 7 is one of the most promising allergens in *A. simplex,* as it is recognized in all infected patients and appears antigenically different from *Pseudoterranova* allergens [86].

Confirmatory and explorative studies on *Anisakis* allergens became available with the advent of omics approaches. In 2016, Baird et al. [92] employed the high-throughput (Illumina) sequencing and bioinformatics comparison of predicted peptides with sequence data available in the AllergenOnline database to identify novel putative allergens among all the molecules of the *A. simplex* and *A. pegreffii* transcriptomes. Cyclophilins, a family of conserved proteins involved in a range of human inflammatory diseases like asthma and rheumatoid arthritis, was observed in both species, and two sequences with high similarity to the nematode polyprotein allergen ABA-1 proteins (members of the nematode polyprotein allergens (NPAs) from *A. suum* and involved in hypersensitivity reactions) were observed in *A. simplex*. Computationally translated amino acid sequences from the two species were also compared with the lists of putative allergens described by Arcos et al. in 2014 [66] through two dimensional gel electrophoresis 2DE gel and MALDI-TOF analyses. Arcos et al. showed differences in the allergenic abilities of the *A. simplex* complex species, and they found that *A. simplex s.s.* showed a higher number of immunoreactive protein spots (34 different spots) than *A. pegreffii* (11 different immunoreactive spots), and both species showed more immunoreactive proteins than their hybrids (six proteins spots).

As mentioned before, the proteomic investigation of *A. simplex* L3s identified 17 novel putative allergens, including structural proteins like myosin-4 and enzymes like enolase and endochitinase [68]. 

The use of high-throughput technologies such as genome sequencing, RNA-seq, and proteomic profiling will enable the characterization of novel allergens and the defining of their functions and properties through sequence epitope prediction and wide comparisons with available databases.

## 6. Future Perspectives

The general picture described in this review indicates an active scientific environment that has been implemented in the last few years (see Box 2). However, steps forward are continuously being undertaken, both by developing new technical approaches and by addressing new topics and aspects. 

In particular, although secreted proteins play a significant role in host manipulation, they represent only one class of the potential actors in the interactions between parasites and hosts. Recent studies of non-coding RNAs (ncRNAs) led to the identification of several classes of ncRNAs involved in a complex regulatory network that modulates gene expression, with particular interest on microRNAs (miRNAs). MiRNAs act on endogenous genes, and pathogens may exploit miRNAs to target host gene expression [93]. This is known for viruses, and it has more recently described for parasites [94,95]. MiRNAs are present into biofluids, often in a protected state within a class of extracellular vesicles named exosomes that also carry other RNAs, as well as proteins and lipids. Within exosomes, miRNAs display selective profiles that differ from the global miRNA contents of the parent cell or tissue, suggesting a modulation role exerted by targeting specific cells or cell-contents. Currently, the miRBase registry contains a substantial number of miRNAs from parasitic helminths (e.g., *Ascaris*), but studies on exosomal miRNAs from parasitic nematodes are at initial stages. These have been conducted on nematodes (*Brugia*, *Trichuris*, and *Heligmosomoides*) and on flatworms such as *Echinococcus* and *Schistosoma*, all suggesting a role in host gene regulation, host–parasite interactions, and host immune evasion.

Studies on miRNAs potentially secreted by *Anisakis* larvae in exosomes may provide their characterization and assist in the identification of potentially targeted human genes, shedding some light on the mechanism of host manipulation by anisakid nematodes. The identification of potential targets could promote the development of highly specific diagnostic tools for parasite-driven allergies [96].

Another interesting topic is the *Anisakis* tumorigenic potential, as reports of cancer co-occurrence with L3 are increasing [4,6,97], and one study suggested previous exposure to *Anisakis* as a risk factor for gastric or colon adenocarcinoma [98]. Moreover, helminth EVs have immunomodulatory effects and may promote tumorigenesis, as demonstrated for parasitic flatworms [99]. Hrabar et al. [100] investigated the interplay between proinflammatory cytokines, miRNAs, and tissue lesions in *Anisakis*-infected Sprague Dawley rats. Though they did not detected differences in DNA methylation between infected and uninfected tissues, the authors advocated for further studies to investigate the tumorigenic potential of these nematodes. Similar aspects have been investigated using in vitro models as human fibroblasts [101], human dendritic cells [102], and human epithelial colonic cancer cells [103], revealing modulatory activity exerted by *Anisakis* products, such as the upregulation of oxidative-stress, the inhibition of apoptosis-related biomarkers, and inflammatory induction. The tumorigenic potential of *Anisakis* has been recently explored using hamster ovary cells and Sprague Dawley rats, revealing an increased cell proliferation, decreased apoptosis, and changes in the expression of serum cancer-related miRNAs in rats [29].

As already reported in both model and non-model organisms, advances in genomics and transcriptomics have boosted the development of functional studies by reverse genetics approaches. In parasitology, these studies can contribute to improving the basic knowledge of parasitic pathogenesis, metabolic pathways, life-cycle development, host–parasite interactions, etc., and, importantly, to opening new perspectives in the discovery of new drug targets [104]. Here, we highlight a few key concepts related to the most important tools currently available in the study of gene function in parasitic nematodes, e.g., reverse genetics by RNA interference and genome editing by CRISPR-Cas9 (Clustered Regularly Interspaced Short Palindromic Repeats and Caspase9 technology) [105,106]. Since the discovery of specific gene silencing by RNA interference (RNAi) in *C. elegans*, several efforts have been made to develop transient gene silencing protocols in parasitic nematodes. Initially, the application of RNAi technology to parasitic nematodes was not as straightforward as expected, and efficacy has been extremely variable [107]. Among the main critical issues encountered, it is worth mentioning the lack of appropriate culture conditions, the implementation of methods of delivery, and the choice of parasite stage [107,108]. However, there have been successful examples of anisakids cultivation [109,110]. 

The recent availability of high throughput molecular tools has provided a novel opportunity to study gene function at several levels [111]. Thus far, most of the effective reverse genetics tools mediated by RNAi are available in three commonly studied nematodes: *Strongyloides* spp., *Brugia malayi*, and *Ascaris suum* [106,111,112]. The development of RNAi protocols in other parasitic nematodes of medical and veterinary importance, e.g., *Anisakis* spp., would offer novel opportunities for both basic research and for the identification and validation of drug targets.

Box 2Highlights from *Anisakis* omics studies (see also Table 2).**GENOMES:** Currently, two genomes from *A. simplex* L3 are available in public repositories. They count around 125 million bp and around 21 thousand predicted genes. The first topic investigated was the carbohydrate metabolism in *A. simplex*
*s.l.* L3 and L4, with particular attention to glycogen and trehalose. **TRANSCRIPTOMES:**
Transcriptome studies have been performed on (i) the entirety of L3 to obtain a wider catalogue of potential allergens, wherein an additional 35 molecules (from 15 known allergens) were suggested as new allergens; (ii) L3 and the dissected pharyngeal region to obtain a list of tissue-enriched molecules with a potential role in pathogenesis, where a list of peptidases, toxins, and allergens was provided; and (iii) L3 and L4 to obtain molecules specifically enriched in the two developmental stages, where it was found that proteases and collagens were highly expressed in the L3 and L4, respectively. **PROTEOMES:** Studies on the entire set of *Anisakis* proteins assisted the discovery of new allergenic molecules, with particular attention to zoonotic species and their hybrid form. These studies confirmed those allergens involved in cross-reactions through observations of serological assays.

**Table 2 genes-11-00801-t002:** List of available omics studies on anisakids. Genomics, transcriptomics, and proteomics studies on anisakids with information on parasitic species analyzed, developmental stage or tissue, major findings, accession number of available data in public repositories, and references. * AnisakisDB is available at www.anisakis.mncn.csic.es. (WB: Western blot. Public repositories: SRA: Sequence Research Archive. ENA: European Nucleotide Archive. PX: ProteomeXchange).

Omics Approach	Parasitic Species	Developmental Stage and Tissue	Major Findings	Public Repositories	References
Genomics	*Anisakis simplex*	Not available	First draft with genome size, annotation and gene counts	SRA PRJEB496	[22]
Genomics	*Anisakis simplex sensu lato*	L3 L4/6 days L4/12 days L5 (pre-adult) adult	Carbohydrate metabolism during life cycle (trehalose and glycogen metabolism)	ENA ERS2790326	[34]
Transcriptomics	*Anisakis simplex sensu stricto Anisakis pegreffii*	L3	First curated list of transcripts with focus on potential allergens	SRA SRP070744	[92]
Transcriptomics	*Anisakis simplex* *Anisakis pegreffii*	L3 L4	First comparative study on different developmental stages		[43,44]
Transcriptomics	*Anisakis simplex* *sensu stricto* *Anisakis pegreffii*	L3 and dissected pharyngeal region	First comparative study on two anisakid zoonotic species and their tissue-specific molecules	SRA PRJNA374530	[33]
Transcriptomics	*Anisakis simplex* *sensu stricto* *Anisakis pegreffii* *Hysterothylacium aduncum*	L3 and dissected pharyngeal region	First study on a raphidascaridid species	SRA PRJNA601087	[48]
Transcriptomics	*Anisakis simplex* *sensu stricto* *Anisakis pegreffii* and their hybrids	L3	First study on hybrids	SRA SRP072976and AnisakisDB *	[60]
Proteomics	*Anisakis simplex sensu strictoAnisakis pegreffii* and their hybrids	L3, mass spectrometry and WB with sera from allergic patients	Description of novel potential allergens	PX PXD000662	[66]
Proteomics	*Anisakis simplex*	L3 and sera from sensitized patients and mass spectrometry	Allergens with focus on cross reactivity		[68]
Proteomics	*Anisakis* sp. *Pseudoterranova* sp.		Protein biomarker discovery and fast monitoring to identify and detect Anisakids in fishery products		[70]
Proteomics	*Anisakis simplex*	L3 L4	First global proteomes of *A. simplex* L3 and L4		[71]

## Figures and Tables

**Figure 1 genes-11-00801-f001:**
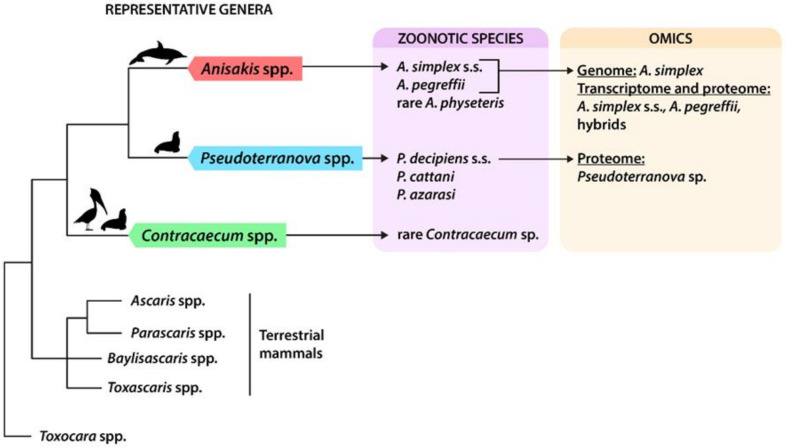
Simplified phylogeny of anisakids, with indications of zoonotic species and available omics data. Simplified phylogenetic relationships of anisakids, modified from the last updated inference based on the entire mitochondrial (mtDNA) [11] can be seen in Figure 1; other phylogenetically related ascaridoids of terrestrial mammals are included as sister groups. Indications of final host affiliation for anisakids are reported above the branches (cetaceans, pinnipeds, and fish-eating birds). References of omic studies are reported in the text and in Table 2.

**Table 1 genes-11-00801-t001:** List of identified *Anisakis simplex* allergens, with indications of codes according to the allergen database (AllFam), the UniProt database, and the nature of protein (E/S: excreted/secreted; S: somatic), along with references to the literature.

Allergen (kDa)	AllFam Family ID	Description	UNIPROT	Location of the Products	References
Ani s 1 (24)	AF003	Animal Kunitz serine protease inhibitor	Q7Z1K3	E/S major allergen	[78]
Ani s 2 (97)	AF100	Myosin heavy chain (paramyosin)	Q9NJA9	S panallergen/major allergen	[79]
Ani s 3 (41)	AF054	Tropomyosin	Q9NAS5	S panallergen/major allergen	[81]
Ani s 4 (9)	AF005	Cystatin	Q14QT4	E/S	[82]
Ani s 5 (15)	AF137	SXP/RAL-2 family	A1IKL2	E/S	[83]
Ani s 6 (7)	AF027	Cysteine-rich trypsin inhibitor-like domain	A1IKL3	E/S major allergen	[83]
Ani s 7 (139)	Unclass	Armadillo ARM-like	A9XBJ8	E/S major allergen	[81]
Ani s 8 (15)	AF137	SXP/RAL-2 family	A7M6Q6	E/S	[83]
Ani s 9 (14)	AF137	SXP/RAL-2 family	B2XCP1	E/S	[73]
Ani s 10 (22)	Unclass	Ani s 10 allergen	D2K835	S ?	[88]
Ani s 11 (55)	Unclass	Ani s 11 allergen	E9RFF3	S ?	[89]
Ani s 11-like (?)	Unclass	Ani s 11 allergen	E9RFF5	S ?	[89]
Ani s 12 (?)	Unclass	Ani s 12 allergen	L7V3P8/E9RFF6	major allergen	[89]
Ani s 13 (36.7)	AF009	Globin	A0A1W7HP35	?	[90]
Ani s 14 (23.5)	Unclass	ARM-like	A9XBJ8	?	[91]

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
