# Peer review of "Advances in Omic Studies Drive Discoveries in the Biology of Anisakid Nematodes"

_genes, 2020, doi:10.3390/genes11070801_

Round 1

Reviewer 1 Report

Overall, this review seems a bit lengthy and unfocused.  From the title, I would expect a review on recent omics studies, but the authors spend considerable time discussing other studies that seem tangential.  The addition of a few figures and tables would help to summarize key points.  A revised manuscript has potential to be helpful and informative for the field.

Major Points:

A summary figure of Anasakid species, with their phylogeny as well as notes on the status of genome/transcriptome/proteome information, would be helpful.  Notes on the definitive host(s) might also be helpful, as well as species known to be human pathogens.

A summary table of genomic, transcriptomic, and proteomic studies, along with relevant tissues/developmental stages examined, would be very helpful.  Also, it would be helpful to list relevant databases.

There are several paragraphs dedicated to previous work that, while important, does not seem to be directly related to the topic of the review - that is “advances in omic studies.”  A review that is more focused would be more useful for the reader.  For example, the section on transcriptomics contains several paragraphs of studies (e.g. lines 196-209) not involving transcriptomics; similarly, the section on proteomics details several studies not involving proteomics.  There is an entire section (lines 288-375) in “Characterization of allergens” that does not mention any omics studies (this information is summarized in Table 1).

Minor Points:

Line 25 (also on line 28): I would recommend replacing “squids” with “cephalopods” (see Abollo et al. 2001 Parasitology Research).

Lines 26-27: It might be helpful to note that eggs passed in the feces then embryonate in the seawater, and that the larvae moult twice inside the egg.  Thus it is the third-stage larva that is infectious for the paratenic host.

Line 28: Note that it is the third-stage larva that bores through the digestive tract wall.

Line 32: Also note anisakiasis (this is used elsewhere in the text).

Line 34: Describe the allergen aspect a bit more clearly.

Line 46: Maybe also briefly mention how anasakids are phylogenetically/evolutionarily related to other parasitic nematode species?  See Clade 8B nematodes in van Megen et al. 2009 Nematology.

Lines 67-68: The statement “since parasitic helminths generally show much more complex genomes than their closely related models or free-living organisms” is not entirely correct.  There are several clades of parasitic nematodes (e.g. Clades I and IV, see Blaxter et al. 1998) that have more compact genomes and fewer genes than the free-living model nematodes (there is a table illustrating this in Foth et al. 2004 Nature Genetics, as well as a supplemental table in the “Comparative genomics of the major parasitic worms” paper cited later).  References 15 & 16 do not appear to be correct.

Line 73: There are a few more up-to-date reference for reviews on RNA-Seq and transcriptomics in parasitic nematodes.

Line 88 : A summary figure with Box 1 would be helpful.

Line 109: Should be “WormBase ParaSite” which is different from WormBase (this should also be corrected elsewhere in the manuscript).

Line 112: Spell out “s.l.” first time used.

Line 116: Why are glycogen and trehalose the most important saccharides?

Lines 109 - 120: Some of this comparison information might be better summarized in a table.

Lines 148-149: Caveat: mitochondrial genome phylogenies do not always agree with nuclear genome phylogenies (see example in Hunt et al. 2016).

Line 176: What type of proteins do A.peg genes encode?

Line 181: Use “Carbohydrate” instead of “Sugar”?  Also, it is unclear in this paragraph whether the parasite is catabolizing carbohydrates for energy or synthesizing them for other purposes.

Lines 187-195: Unclear whether temperature requirements are simply a result of parasite co-evolution with ectothermic hosts.

Lines 196-202: Unclear what the conditions were for the enzymatic studies (i.e. what two conditions are being compared)?  For this and the following paragraph, I’m not sure how these enzymatic studies relate to recent advances in ‘omics.

Lines 210-238: This paragraph is unfocused and has several topics.  It needs to be broken up.

Lines 215-217: If this is transcriptomic study, then the molecules detected would be transcripts that encode various predicted proteins, not the proteins themselves (as it is written).  This needs to be clarified here, as well as other places in the manuscript.

Line 227: I’m not sure that “tardive” is the word you want to use here.

Line 233: Should be written “transcripts encoding metalloproteinases”

Line 256: The number of hits for the search term “Anisakis” in NCBI should not be conflated with the number of genes (some proteins are represented multiple times, etc.).  This does not appear to be appropriate data collection.

Line 266: De novo sequencing is not a proteomics method (the section is about proteomics).

Lines 272-274: I am unsure how sequence homology between related groups of proteins equates with cross-reactivity.

Lines 274-276: Presence of allergen-like proteins and their expansion in parasitic nematode lineages has been described in other papers (e.g. SCP/TAPS as reviewed in the “Comparative genomics of the major parasitic worms” paper cited previously).

Lines 288-296: Are there two or three forms of allergic anisakiasis?  Also, anisakidosis vs. anisakiasis is not used consistently throughout.

Lines 288-375: In this section on “Characterization of allergens” - if there haven’t been any recent studies using omics to study allergens, why are these other studies are reviewed here?  This section, while informative and detailed, seems out of place in this review and would seemingly be better suited for a different review article.

Lines 315-325: Describe how paramyosin and tropomyosin, which are highly conserved muscle proteins, can be allergens.  Some readers will be skeptical of this.

Line 336: Do recently sequenced genomes have genes encoding protein with homology to Ani s 7?  Is it unique to anisakids?

Line 378: It is unclear how the novel putative allergens were identified (methodologically).

Line 379: Species are not typically abbreviated in this way, nor are they abbreviated elsewhere in the text.

Lines 389-393: This paragraph does not seem to entirely fit in this section.  A mention of the Anisakis Data Base before line 393 seems warranted.

Line 404-418: Future perspectives paragraph is lacking references.  Just a few examples: Line 406: the “recent studies” are not cited, Line 410: the miRNAs in parasites is lacking a citation, etc.

Line 419: Qualify statement about Anisakis exosomes, unless they are known to secrete these (and cite the paper).

Line 426: Abbreviation “EV” is not used elsewhere.  Spell out.

Line 456: Has anyone tried to do RNAi in Anisakis spp.?

Line 515: The release number needs to be stated.

Writing:

There are multiple places in the text with errors in English grammar, syntax, and sentence structure.  I would strongly recommend having this manuscript proof-read by a copy editor prior to publication.  I think it would also be helpful to have a writing coach assist with ways to condense some of the writing, which is a bit verbose in places.  Stylistically, it is also readily apparent that different sections were composed by different individuals.

Author Response

Reviewer 1

Overall, this review seems a bit lengthy and unfocused.  From the title, I would expect a review on recent omics studies, but the authors spend considerable time discussing other studies that seem tangential.  The addition of a few figures and tables would help to summarize key points. A revised manuscript has potential to be helpful and informative for the field.

Reply: We greatly appreciated the reviewer’s effort to revise in details the manuscript, suggesting also to include figures and tables, that we are sure will improve it. We have prepared a figure with Anisakis species and other anisakids, and a Table summarizing the omics study on the field. Changes are all tracked by the revision system in Words and given the modification in Reference numbered list we have replaced it with the updated list of citations. Line numbers cited along this rebuttal referred to the original version of the submitted manuscript (as originally received in the revision).

Major Points:

A summary figure of Anasakid species, with their phylogeny as well as notes on the status of genome/transcriptome/proteome information, would be helpful.  Notes on the definitive host(s) might also be helpful, as well as species known to be human pathogens.

Reply: A figure with a simplified phylogenetic picture of anisakids and other members of the superfamily ascaridoidea has been included, together with info on final hosts, indication of zoonotic species and on omic studies. The figure has been related to the Box1.

A summary table of genomic, transcriptomic, and proteomic studies, along with relevant tissues/developmental stages examined, would be very helpful.  Also, it would be helpful to list relevant databases. Regarding genome/transcriptome/proteome information, we have followed the suggestion and a Table have been included, in relation to box2 in the text.

There are several paragraphs dedicated to previous work that, while important, does not seem to be directly related to the topic of the review - that is “advances in omic studies.”  A review that is more focused would be more useful for the reader.  For example, the section on transcriptomics contains several paragraphs of studies (e.g. lines 196-209) not involving transcriptomics; similarly, the section on proteomics details several studies not involving proteomics.  There is an entire section (lines 288-375) in “Characterization of allergens” that does not mention any omics studies (this information is summarized in Table 1).

Reply: The section on transcriptomics has been shortened according to the suggestion of the reviewer. As for the section on allergens, we agree with the reviewer that it is only marginally consistent with the core of the paper. However, we believe it is worthy to keep a shorter version of this paragraph to provide a detailed list of the general information on known allergens so far available and to underline the lack of knowledge in this context, whether or not related to omics. Both sections has been revised and reduced in length.

Minor Points:

Line 25 (also on line 28): I would recommend replacing “squids” with “cephalopods” (see Abollo et al. 2001 Parasitology Research). Reply: the terms have been replaced.

Lines 26-27: It might be helpful to note that eggs passed in the feces then embryonate in the seawater, and that the larvae moult twice inside the egg.  Thus it is the third-stage larva that is infectious for the paratenic host. Reply: the sentence “with third-stage larvae (L3)” have been added.

Line 28: Note that it is the third-stage larva that bores through the digestive tract wall. Reply: yes we agree, after adding the previous sentence it is more clear.

Line 32: Also note anisakiasis (this is used elsewhere in the text). Reply: the term anisakiasis has been added.

Line 34: Describe the allergen aspect a bit more clearly. Reply: We have mentioned “as rhinitis, urticaria and anaphylactic shock”

Line 46: Maybe also briefly mention how anasakids are phylogenetically/evolutionarily related to other parasitic nematode species?  See Clade 8B nematodes in van Megen et al. 2009 Nematology. Reply: the sentence has been revised “The evolutionary relationships display a complex scenario, where marine Raphidascarididae appear more closely related to terrestrial Ascarididae than to Anisakidae, the latter being paraphyletic” together with the suggested citation.

Lines 67-68: The statement “since parasitic helminths generally show much more complex genomes than their closely related models or free-living organisms” is not entirely correct.  There are several clades of parasitic nematodes (e.g. Clades I and IV, see Blaxter et al. 1998) that have more compact genomes and fewer genes than the free-living model nematodes (there is a table illustrating this in Foth et al. 2004 Nature Genetics, as well as a supplemental table in the “Comparative genomics of the major parasitic worms” paper cited later).  References 15 & 16 do not appear to be correct. Reply: According to the reviewer’s suggestions, the statement has been changed as follows:” Indeed, there are examples where the genome of parasitic helminths is more complex than that of their closely related models or free-living organisms [18-20].” References have been updated accordingly.

Line 73: There are a few more up-to-date reference for reviews on RNA-Seq and transcriptomics in parasitic nematodes. Reply: we have included also Elsworth B, Wasmuth J, Blaxter M. NEMBASE4: the nematode transcriptome resource. Int J Parasitol. 2011;41(8):881-894. doi:10.1016/j.ijpara.2011.03.009

Line 88 : A summary figure with Box 1 would be helpful. Reply: we have included reference to Figure 1.

Line 109: Should be “WormBase ParaSite” which is different from WormBase (this should also be corrected elsewhere in the manuscript). Reply: we have added ParaSite along the manuscript.

Line 112: Spell out “s.l.” first time used. Reply: sensu lato has been added.

Line 116: Why are glycogen and trehalose the most important saccharides? Reply:  It is important for molting process, as consecutive development stages occurring into the same host (final host) or for survival, when exposed to thermal, oxidative and desiccation stress, nematodes use to regulate carbohydrate metabolism to ensure high survival. The following sentence has been added: “being involved in molting process and response to stress”

Lines 109 - 120: Some of this comparison information might be better summarized in a table. Reply: as suggested previously by the reviewer, we have included a comprehensive Table (Table 2) with a complete list of omics approach on anisakids and their major findings.

Lines 148-149: Caveat: mitochondrial genome phylogenies do not always agree with nuclear genome phylogenies (see example in Hunt et al. 2016). Reply: Markers with different evolutionary rates may of course generate different phylogenetic signals and we are aware of several examples of contradictory outcomes. In this particular case accounting for Anisakids, the topology of trees testing these evolutionary relationships agreed. In the future, phylogenomics will support further and more reliable phylogeny.

Line 176: What type of proteins do A.peg genes encode? Reply: we have included the protein nature along with antigens codes.

Line 181: Use “Carbohydrate” instead of “Sugar”?  Also, it is unclear in this paragraph whether the parasite is catabolizing carbohydrates for energy or synthesizing them for other purposes. Reply: we have changed sugar with Carbohydrate. The role of such molecules were previously specified in lines 119-120.

Lines 187-195: Unclear whether temperature requirements are simply a result of parasite co-evolution with ectothermic hosts. Reply: There are no clear evidences of host-parasite coevolution related to temperature. The final hosts for Anisakids and Hysterothylacium are different (marine mammals vs teleosts) as their body temperatures are, and consequently co-adaptive processes may explain why Hysterothylacium is not highly pathogenic for humans (just because it is not able to survive at human body temperature).

Lines 196-202: Unclear what the conditions were for the enzymatic studies (i.e. what two conditions are being compared)? For this and the following paragraph, I’m not sure how these enzymatic studies relate to recent advances in ‘omics. Reply: we agreed and we have shortened the paragraph as follows: “Previous studies investigated the role of hydrolases in Hysterothylacium aduncum using classic approaches reporting no trypsin- and chymotrypsin-specific activities as well as alpha-galactosidase, alpha-mannosidase, and beta-glucuronidase thus suggesting a main role of interaction with environmental host rather than the taxonomic affiliation [50].”

Lines 210-238: This paragraph is unfocused and has several topics.  It needs to be broken up. Reply: the paragraph is focused mainly in potentially pathogenic molecules found enriched in L3 or in its specific tissue. We have indeed modified the paragraph by removing the following sentence not directly referred to omic studies: Paragraph removed:  “The same anatomical region was recently investigated by Mladineo et al. [48] using a functional ultrastructure approach on excretory gland cells on larval A. pegreffii and adult Pseudoterranova azarasi. The study suggested anaerobic glycolysis as the main metabolic pathway, achieved through nutrient endocytosis.”)

Lines 215-217: If this is transcriptomic study, then the molecules detected would be transcripts that encode various predicted proteins, not the proteins themselves (as it is written).  This needs to be clarified here, as well as other places in the manuscript. Reply: We have modified accordingly see line 225

Line 227: I’m not sure that “tardive” is the word you want to use here. Reply: changed tardive to secondary

Line 233: Should be written “transcripts encoding metalloproteinases” Reply: done.

Line 256: The number of hits for the search term “Anisakis” in NCBI should not be conflated with the number of genes (some proteins are represented multiple times, etc.).  This does not appear to be appropriate data collection. Reply: we agree that potential redundancy and possible missing of some proteins make these numbers not fully reliable so the sentence has been removed.

Line 266: De novo sequencing is not a proteomics method (the section is about proteomics). Reply: we thanks the reviewer and the sentence has been deleted.

Lines 272-274: I am unsure how sequence homology between related groups of proteins equates with cross-reactivity. Reply: the sequence homology among proteins, in particular in epitopes regions, may be related to cross-reactivity but of course is not a paradigm. The sentence relates to what reported by Faeste et al. However the sentence have been mitigated as follows “which may be the source of possible cross-reactivity.

Lines 274-276: Presence of allergen-like proteins and their expansion in parasitic nematode lineages has been described in other papers (e.g. SCP/TAPS as reviewed in the “Comparative genomics of the major parasitic worms” paper cited previously). Reply: we have added the following sentence “as confirmed also by comparative genomic approaches [69]”.

Lines 288-296: Are there two or three forms of allergic anisakiasis?  Also, anisakidosis vs. anisakiasis is not used consistently throughout. Reply: we have changed two to three and we have modified the term according to current alternative use of anisakiasis and anisakidosis along the manuscript.

Lines 288-375: In this section on “Characterization of allergens” - if there haven’t been any recent studies using omics to study allergens, why are these other studies are reviewed here?  This section, while informative and detailed, seems out of place in this review and would seemingly be better suited for a different review article. Reply: we agree with the reviewer that only one study using transcriptomic approach to explore this topic is available so far, but in our opinion the topic is of great interest. In fact, in a review aimed also to emphasize the more interesting aspect that deserves to be explored in the next future in the framework of anisakids biology, a comprehensive list of allergens and their features can be of help for the readers. However, as stated above, the entire section has been revised and reduced.

Lines 315-325: Describe how paramyosin and tropomyosin, which are highly conserved muscle proteins, can be allergens. Some readers will be skeptical of this. Reply: paramyosin and tropomyosin are well known helminthic molecules that can trigger immune response in humans (in nematodes and trematodes several studies were carried out) and they have been deepen for their ability to bind IgE. In the framework of anisakids, some studies were performed on native, cloned and recombinant Ani s 2 and Ani s 3 allergens with libraries to explore specific IgE binding protein antigens, thus confirming their allergenic role (see citations 79-81 in the manuscript).

Line 336: Do recently sequenced genomes have genes encoding protein with homology to Ani s 7? Is it unique to anisakids? Reply: We are not aware of such homologies. A BLASTn search using the available nucleotide sequence of Ani s 7 did not reveaI homology with any other organism. A protein search using aminoacidis residues revealed very low identity with hypothetical protein of T. canis (33.55% KHN86688).

Line 378: It is unclear how the novel putative allergens were identified (methodologically). Reply: we have added these details “bioinformatics comparison of predicted peptides with sequence data available in the AllergenOnline database”

Line 379: Species are not typically abbreviated in this way, nor are they abbreviated elsewhere in the text. Reply: we have removed abbreviations AS and AP

Lines 389-393: This paragraph does not seem to entirely fit in this section.  A mention of the Anisakis Data Base before line 393 seems warranted. Reply: as suggested by the reviewer, the paragraph has been moved to the end of the Transcriptome chapter.

Line 404-418: Future perspectives paragraph is lacking references. Just a few examples: Line 406: the “recent studies” are not cited, Line 410: the miRNAs in parasites is lacking a citation, etc. Reply: we thanks the reviewer to have notified it and the following references has been included:

Kim, V. N. MicroRNA biogenesis: coordinated cropping and dicing. Nat Rev Mol Cell Biol 6, 376–385, https://doi.org/10.1038/nrm1644 (2005).

Bartel, D. P. Metazoan MicroRNAs. Cell 173, 20–51, https://doi.org/10.1016/j.cell.2018.03.006 (2018).

Zheng Y, Cai X, Bradley JE. microRNAs in parasites and parasite infection. RNA Biol. 2013;10(3):371-379. doi:10.4161/rna.23716

Line 419: Qualify statement about Anisakis exosomes, unless they are known to secrete these (and cite the paper). Reply: Given the publications of EVs produced by other helminthic parasites and in particular by other gastrointestinal nematodes (i.e. Ascaris), is supposed that also Anisakis produce EV. In fact it actually produces EVs (because we are studying them, not yet published paper). However, we used conditional term “eventually” and “may provide”.

Line 426: Abbreviation “EV” is not used elsewhere.  Spell out. Reply: the abbreviation was firstly used at the initial part paragraph of the proteomic chapter.

Line 456: Has anyone tried to do RNAi in Anisakis spp.? Reply: to our knowledge not yet. However, it is likely that this tool will be used in the near future to infer/confirm pathways and gene functions in anisakids as already done in other parasitic nematodes.

Line 515: The release number needs to be stated. Reply: “release 14, 2019” has been added.

Writing: There are multiple places in the text with errors in English grammar, syntax, and sentence structure I would strongly recommend having this manuscript proof-read by a copy editor prior to publication. I think it would also be helpful to have a writing coach assist with ways to condense some of the writing, which is a bit verbose in places. Stylistically, it is also readily apparent that different sections were composed by different individuals. Reply: the manuscript has been throughout checked for fixing errors and English grammars.

Reviewer 2 Report

In the reviewer's opinion, this article will be well received by researchers working on the subject of this review as it is an update from a current and future perspective not covered by other reviews. It's interesting and it fills a gap in anisakiasis reviews.

The article is well structured and presented correctly. Some minor corrections to the manuscript have been suggested as comments in the pdf file.

Also, some comments have been made in the archive that we hope can be useful to the authors.

The authors should be aware of a paragraph in section 6 "Future perspectives" (lines 404-418) where data are provided but it seems that the references have been lost.

Author Response

Reviewer 2

In the reviewer's opinion, this article will be well received by researchers working on the subject of this review as it is an update from a current and future perspective not covered by other reviews. It's interesting and it fills a gap in anisakiasis reviews. The article is well structured and presented correctly. Some minor corrections to the manuscript have been suggested as comments in the pdf file. Also, some comments have been made in the archive that we hope can be useful to the authors. The authors should be aware of a paragraph in section 6 "Future perspectives" (lines 404-418) where data are provided but it seems that the references have been lost.

Reply: The authors thanks the reviewer 2 for the useful comments and modifications suggested, that we have all addressed to our best. Missing references have been included in the first paragraphs of the Future perspectives chapter as well. Line numbers referred to the original version of the submitted manuscript. Please find below a detailed rebuttal to comments included in the pdf:

line43: Why in capital letters? Anisakids should be written with a lowercase. If the authors prefer capital letters they should write Anisakidae in reference to this family of nematodes. Please apply to the rest of the text. Reply: we have changed Anisakids to anisakids along the manuscript.

Line 121 Both 'sensu stricto' and 'sensu lato' are Latin expressions. For this reason, they should be written in italics.Please apply to the rest of the text. Reply: we have used italics along the manuscript.

Line 260 As in the first comment about the word  "anisakids", "anisakiasis" should also be written in lowercase. Reply: we have changed capital letter to lowercase along the manuscript.

Line261 The authors should be more precise and reduce this area to the countries of the northern basin of the Western Mediterranean Sea until more studies are conducted. Similarly, among far eastern countries, only Anisakis pegreffii has been shown to be predominant in Yellow Sea and southern Japan Sea (South Korea and western Japan). Reply: we have changed the sentence as follows “European Mediterranean areas as Italy, Spain and Croatia and also in eastern countries as Korea and Japan”

Lines 274-275 It is true that there is an ancestral phylogenetic relationship between these two groups of living beings, but there is no doubt that they have diverged significantly. The reviewer does not think it is necessary to resort to this explanation because then the innumerable cases of convergent evolution between living beings that are very far apart phylogenetically and that reach similar solutions to similar problems are forgotten.  Reply: as suggested by the reviewer we have accounted also the hypothesis of convergent evolution and modified the sentence as follows: “This is a limiting factor for diagnostic specificity as common phylogenetic route or convergent evolution may lead to common protein solution for similar pathway (the ability to moult shared by arthropod and nematode, both belonging to the Ecdysozoa superphylum).”

Lines 306-307 Allergens as Ani s 1 and Anis s 10 are also heat resistant. The reference [68] is only for Ani s 9. And references for other data in the text? Reply: thanks to the reviewer for the comment. We have cited also the two allergens missing and we have added references. Carballeda-Sangiao N et al J Food Prot. 2014;77(4):605-609 and Carballeda-Sangiao N et al Int Arch Allergy Immunol. 2016;169(2):108-112.

Line326 This data is for the recombinant form. The native form appears to have 9 kDa (Moneo et al. 2005, Parasitol Res. 96:285–9) Reply: we thanks the reviewer and we have changed text and added the reference.

Line 334: we have included inhibitor as suggested, thanks for the comment.

Line 348: Kobayashi et al.'s (your ref. 79) "data demonstrated the IgE cross-reactivity between Ani s 8 and Ani s 5". Please, re-read the reference [68]. Reply: we thanks the reviewer and we have changed the sentence reporting the cross reactivity, moving also the references.

Line 385: we have added et al after Arcos

Line 386: we have changed simplex to italics

Lines 404-417 Have the references been missed? There is a lot of information in this paragraph that needs to be referenced. Without references, the interested reader will not be able to resort to the original sources. Reply: as already mentioned in the reviewer 1 comment, we thanks the reviewers to have notified it and the following references has been included:

Kim, V. N. MicroRNA biogenesis: coordinated cropping and dicing. Nat Rev Mol Cell Biol 6, 376–385, https://doi.org/10.1038/nrm1644 (2005).

Bartel, D. P. Metazoan MicroRNAs. Cell 173, 20–51, https://doi.org/10.1016/j.cell.2018.03.006 (2018).

Zheng Y, Cai X, Bradley JE. microRNAs in parasites and parasite infection. RNA Biol. 2013;10(3):371-379. doi:10.4161/rna.23716

Line 428: we have corrected invetigates to investigated

Line 448-449: Although it is true that the cultivation of anisakids is difficult, for many years the achievement of adults and eggs of the most interesting species such as Anisakis simplex s.l., Pseudoterranova decipiens s.l. and Contracaecum osculatum s.l. have been published. For example: Townsley et al. 1963. J. Fish. Res. Board Canada 20, 743–747. https://doi.org/10.1139/f63-049

Grabda. 1976. Acta Ichthyol. Piscat. 6, 119–141. https://www.aiep.pl/volumes/1970/6_1/pdf/6_1_08.pdf

Likely & Burt. 1989. Can. J. Fish. Aquat. Sci. 46, 1095–1096. https://doi.org/10.1139/f89-141

Likely & Burt. 1992. Can. J. Fish. Aquat. Sci. 49, 347–348. https://doi.org/10.1139/f92-039

Iglesias et al. 2001. Parasitology 123, 285–291. https://doi.org/10.1017/S0031182001008423

Moreover, for the reviewer, the great problem of this type of studies that include cultivation is the exact knowledge of the species. Which can only be achieved by molecular diagnosis, but the parasite is destroyed. It is therefore necessary to develop some technique that allows the identification in vivo of the species under study. Reply: we thanks the reviewer for the interesting citations on anisakids cultivation and we have added a sentence with two selected references. “However, successful examples of anisakids cultivation are available”.

Round 2

Reviewer 1 Report

Overall, I think that the addition of Figure 1 and Table 2 definitely contribute to this review.  I still think Section 5 (allergens) could use a concluding paragraph describing how high-throughput technologies will allow for characterization of allergen function, potential treatments for patients, and identification of additional allergens of possible medical importance.

A few minor points:

Line 26 of the revised MS: definitive host is removed from line 26, but present on line 31.  Be consistent.

Line 33 of the revised MS: replace “due to” with “which results from”

Lines 8-9 and line 66 of the revised MS: unclear why NGS was removed. For nucleic acid-based studies, most current studies are NGS-based. It is also a key term that people will search for.  Ultimately, it is the author’s choice as to whether to include it.

Line 128 of the revised MS: “instance”

Lines 139-144 of the revised MS: It would be helpful if the comparisons to other parasitic nematodes were consistent.  Some comparisons are to Toxocara, others to Brugia, and another to Haemonchus.

Line 178 of the revised MS: sentence ending here is missing a reference.

Line 228 of the revised MS: hydrolases are enzymes; thus, they cannot have a role in “taxonomic affiliation” as written.

Line 482 of the revised MS: should be “potentially” rather than “eventually.”

Writing:

There are still multiple places in the text with errors in English grammar, syntax, and sentence structure.  I have listed three examples below; there are multiple others.  I would strongly recommend having this manuscript proof-read by a professional copy editor prior to publication.

Line 161 of the revised MS: something seems to be missing in the edits to this sentence.

Lines 167-169 of the revised MS: revisions to this sentence do not make grammatical sense.

Lines 186-188 of the revised MS: this is an incomplete sentence.

Author Response

All the authors would like to thank the reviewer 1 for the further efforts made in the revision of the manuscript, that will certainly contribute to its improvement.

A detailed rebuttal has been attached.  
